# Analysis of the Distribution Characteristics of Jellyfish and Environmental Factors in the Seawater Intake Area of the Haiyang Nuclear Power Plant in China

**DOI:** 10.3390/biology13060433

**Published:** 2024-06-12

**Authors:** Yunpeng Song, Tiantian Wang, Minsi Xiong, Shenglong Yang, Heng Zhang, Jie Ying, Yongchuang Shi, Guoqing Zhao, Xiumei Zhang, Xiaodan Liu, Cankun Lin, Zuli Wu, Yumei Wu

**Affiliations:** 1East China Sea Fisheries Research Institute, Chinese Academy of Fishery Sciences, Shanghai 200090, China; syp9805@163.com (Y.S.); xiongms@ecsf.ac.cn (M.X.); ysl6782195@126.com (S.Y.); zhangziqian0601@163.com (H.Z.); shiyc@ecsf.ac.cn (Y.S.); zgq617717@163.com (G.Z.); 2College of Marine Living Resource Sciences and Management, Shanghai Ocean University, Shanghai 201306, China; 3Key Laboratory of Fisheries Remote Sensing, Ministry of Agriculture and Rural Affairs, Shanghai 200090, China; 4Yantai Marine Economic Research Institute, Yantai 264003, China; wangtt2009@163.com (T.W.); xmzhang50542@126.com (X.Z.); 18506382659@163.com (X.L.); 5Zhoushan Yuanjie Aquatic Seed Farm, Zhoushan 316111, China; yingjie3020@163.com; 6School of Marine Information Engineering, Jimei University, Xiamen 361021, China; 15710684756@163.com

**Keywords:** jellyfish, generalized additive model, spatiotemporal distribution, environmental factors, cold source blockade

## Abstract

**Simple Summary:**

Against the background of frequent threats to the cooling water intake systems of nuclear power plants by marine organisms, this study utilized a Generalized Additive Model to investigate the correlation between jellyfish aggregations and environmental factors in the South China Sea region of the Shandong Peninsula. The results indicate that key variables affecting jellyfish resource density include year, longitude, latitude, sea surface temperature (SST), and sea surface salinity (SSS). Subsequently, the study also examined the impact of sea winds and currents on jellyfish resource density. The results suggest that the environmental conditions around marine nuclear power plants (SST, SSS, and ocean current) provide a favorable environment for jellyfish survival.

**Abstract:**

In recent years, there have been frequent jellyfish outbreaks in Chinese coastal waters, significantly impacting the structure, functionality, safety, and economy of nuclear power plant cooling water intake and nearby ecosystems. Therefore, this study focuses on jellyfish outbreaks in Chinese coastal waters, particularly near the Shandong Peninsula. By analyzing jellyfish abundance data, a Generalized Additive Model integrating environmental factors reveals that temperature and salinity greatly influence jellyfish density. The results show variations in jellyfish density among years, with higher densities in coastal areas. The model explains 42.2% of the variance, highlighting the positive correlation between temperature (20–26 °C) and jellyfish density, as well as the impact of salinity (27.5–29‰). Additionally, ocean currents play a significant role in nearshore jellyfish aggregation, with a correlation between ocean currents and site coordinates. This study aims to investigate the relationship between jellyfish blooms and environmental factors. The results obtained from the study provide data support for the prevention and control of blockages in nuclear power plant cooling systems, and provide a data basis for the implementation of monitoring measures in nuclear power plants.

## 1. Introduction

The Shandong Haiyang Nuclear Power Plant is one of the key nuclear power projects in China. The Shandong Haiyang Nuclear Power Plant is located in Haiyang City, Yantai City, Shandong Province. The Haiyang Nuclear Power Plant covers an area of 2256 acres. With a total investment of over one hundred billion yuan, the plan is to build 6 million-kilowatt-level nuclear power units and reserve space for future expansion. Along with the Sanmen Nuclear Power Plant in Zhejiang, the Haiyang Nuclear Power Plant has been designated as a landmark project for the introduction, digestion, and absorption of the third-generation nuclear power technology AP1000 in China. It takes on the important task of achieving China’s nuclear power localization. In recent years, the coastal areas of Haiyang City, Shandong Province, have witnessed recurrent outbreaks and accumulations of drifting marine organisms, such as *Nemopilema nomurai*, *Enteromorpha*, and *Sargassum horneri*, leading to an increased risk of blockage in the cold water intake system of neighboring nuclear power plants [1]. The annual bloom of the green macroalgal *Ulva prolifera* from May to July since 2008 and another of giant jellyfish *Nemopilema nomurai* from June to September have been frequent events in the Yellow Sea [2]. In order to tackle the nuclear safety threat posed by these marine organism outbreaks, assessing and predicting the populations of these organisms in the water intake area of the nuclear power plant is imperative. It bears great significance in ensuring the safe operation of the nuclear power units [3].

The types of blockages in the cooling water intake system of nuclear power plants are influenced by geographical factors and vary across different spatial locations. For instance, coastal nuclear power plants in China and France have experienced abnormal shutdowns caused by blockages from *Aurelia aurita*, *Acaudina molpadioides*, and *Acetes chinensis* [4,5]. Japan had 43 out of 108 thermal power plants affected by jellyfish from 1996 to 2000 [6]. Therefore, when developing marine organism investigation plans, the ecological environment of the site’s sea area should be fully considered, with attention given to the dominant species in the surrounding sea area. Investigation and research on marine organisms should align with local conditions. *R. esculentum Kishinouye* poses significant threats to the cooling water intake system of the Shandong Haiyang Nuclear Power Plant. However, the current monitoring and research on potential disaster-causing organisms such as jellyfish and sand jellyfish in the Haiyang Sea area remain incomplete and have limitations [7]. In the investigation of marine biological resources, scholars both domestically and internationally mainly use high-resolution multibeam sonar detection methods to survey the distribution and density of marine organisms such as jellyfish and krill [7,8,9]. Some scholars have also proposed the establishment and monitoring model of a sonar–optic composite monitoring model and a marine biological monitoring model for the composite monitoring of marine organisms [9,10]. Previous research primarily focused on the abundance and distribution of jellyfish, with limited integration of field surveys and laboratory physical and chemical experiments to analyze changes in jellyfish abundance and the influencing environmental factors.

Given the insufficient early warning capacity of nuclear cold source biological disasters [11], this study is based on the investigation data of jellyfish resources in the southern sea area of the Shandong Peninsula [12,13,14] to assess and analyze the potential blockage risk organisms, jellyfish resources, at the Haiyang Nuclear Power Plant [15]. The aim of this study is to use a Generalized Additive Model (GAM) and SPSS software to analyze the coupling relationship between jellyfish and environmental factors through mathematical statistics. GAM will be applied to quantify the relationship between jellyfish abundance and the environmental variables. SPSS27 software will be used to perform statistical analyses, including regression and correlation analyses, to examine the coupling relationship between jellyfish and environmental factors. Ultimately, this research aims to provide fundamental data for ensuring the safety of the cold water source in the Shandong Haiyang Nuclear Power Plant.

## 2. Materials and Methods

The climate in this area is characterized by a temperate monsoon climate, with four distinct seasons. The summers are warm and humid, while the winters are cold and dry. The ocean currents in the area are mainly influenced by the East Asian monsoon and the warm current of the Yellow Sea. In summer, the Yellow Sea warm current strengthens, leading to higher sea temperatures, while in winter, the prevailing north winds cause a decrease in sea temperatures. In terms of seabed topography, the sea area is characterized by a shallow continental shelf, with water depths generally less than 50 m. The seabed terrain is relatively flat, but there are also some local deep troughs and shallow shoals present. The ocean currents include both coastal currents and transoceanic currents, with noticeable seasonal variations. In summer, the coastal currents flow northward, bringing warmer water from the south, while in winter, they flow southward, carrying colder water from the north. These environmental conditions interact to create a complex hydrodynamic environment in the region.

### 2.1. Jellyfish Resource Survey

The study area for the investigation of jellyfish resources is situated in the coastal waters of the southern Shandong Peninsula between coordinates 36°50′ N–36°50′ N and 121°00′ E–122°20′ E. The survey consists of 15 stations, which are established at regular intervals of 10′ in terms of latitude and longitude [12,16] (Figure 1). According to the nuclear power plant, jellyfish blockages in the cooling system commonly occur from July to September. Therefore, the survey period is from 2010 to 2022, every July of each year, with no data collected in 2020 due to the impact of the COVID-19 pandemic. The survey vessel utilized a flowing net with a main power of 14.7 kW. The jellyfish sampling method employed a specially designed three-layer jellyfish flowing net with a mesh size of 80 mm, a net height of 10 m, and a net length of 1000 m [12,17]. The fishing net was placed at each station for one hour.

The duration of each fishing operation was standardized to one hour. The relative density of jellyfish resources in various years and locations was analyzed using the catch per unit effort (CPUE) index (ind./h). Additionally, a *t*-test was utilized to determine the significance of differences in jellyfish resource density between years and across locations.

### 2.2. Environmental Data

Ocean surface temperature (SST) and sea surface salinity (SSS) data were obtained from the Copernicus Marine Environment Monitoring Service (CMEMS), and the global sea surface significant wave height model data (http://marine.copernicus.eu, accessed on 11 June 2023). The data had a spatial resolution of 0.083° and a temporal resolution of 1 day, containing information such as longitude, latitude, date, temperature (°C), and salinity (‰). The ocean surface wind field data were obtained from CMEMS, which provides a global ocean surface wind observation level 3 product. The wind field data had a horizontal resolution of 0.125° and included information such as longitude, latitude, date, wind speed (m/s), and wind direction (°). The data on ocean currents and wind patterns used in this study were acquired from the GLORYS12V1 dataset provided by CMEMS (https://data.marine.copernicus.eu/product/GLOBAL_MULTIYEAR_PHY_001_030/description, accessed on 11 June 2023). The data had a spatial resolution of 0.083° and a temporal resolution of 1 day. It included essential information such as longitude, latitude, date, flow speed (m/s), and flow direction (°). The time span of the above environmental variable data was from 5 July 2010 to 15 July 2022.

### 2.3. Data Processing Methods

In this study, data collection and processing were conducted using Surfer 13, SPSS Statistics 27, and Matlab R2018a software. Surfer 13 was employed to generate station maps, spatial distribution maps of jellyfish resource density, wind field maps, flow field maps, and so on. Significance tests were conducted on the interannual and spatial differences in jellyfish resource density using SPSS Statistics 27. The normality of jellyfish resource density data within groups was tested using Kolmogorov–Smirnov in SPSS Statistics 27. Pearson correlation analysis was performed on jellyfish abundance, the cumulative effect of ocean currents (the ratio of shoreward current to the total volume of all currents within the monitoring area), site factors, and other variables using SPSS Statistics 27. Matlab R2018a was employed for analyzing the spatial distribution of jellyfish resources using GAM. The environmental data were processed and extracted using MATLAB R2018a. The extracted variables comprised year, month, day, longitude, latitude, sea surface temperature (SST), sea surface salinity (SSS), wind speed (Wind__speed_), the wind direction (Wind__dir_) of the sea surface wind field, sea surface current speed (SSC__speed_), and the sea surface current direction (SSC__dir_). The corresponding environmental data were extracted based on jellyfish survey sites, and the annual mean of each environmental factor during the survey period was calculated.

### 2.4. Generalized Additive Model (GAM)

In this study, GAM [18] was used to analyze the spatial distribution of jellyfish resources in the southern waters of Yantai and Weihai, Shandong Province. In the model, the relative resource density of jellyfish was considered as the response variable [13,19], while the year, longitude, and latitude were considered as temporal and spatial explanatory variables. Environmental explanatory variables, including sea surface temperature (SST), sea surface salinity (SSS), wind speed (Wind__speed_), and wind direction (Wind__dir_) at the sea surface, as well as surface flow speed (SSC__speed_) and flow direction (SSC__dir_), were used [20]. The model was constructed as follows:log(abundance + 1) = *β* + *s*(SST) + *s*(SSS) + *s*(Wind__speed_) + *s*(Wind__dir_) + *s*(SSC__speed_) + *s*(SSC__dir_) + *s*(YEAR) + *s*(LON) + *s*(LAT) + ∈(1)
where abundance is the response variable; *β* is the intercept term; *s*(SST) is the environmental explanatory variable sea surface temperature; *s*(SSS) is the environmental explanatory variable sea surface salinity; *s*(Wind__speed_) is the environmental explanatory variable sea surface wind speed; *s*(Wind__dir_) is the environmental explanatory variable sea surface wind direction; *s*(SSC__speed_) is the environmental explanatory variable sea surface current speed; *s*(SSC__dir_) is the environmental explanatory variable sea surface current direction; *s*(YEAR) is the spatiotemporal explanatory variable year; *s*(LON) is the spatiotemporal explanatory variable longitude; *s*(LAT) is the spatiotemporal explanatory variable latitude; and ∈ is the random error term.

To avoid multicollinearity among environmental variables, the relationships between these variables were examined using the Variance Inflation Factor (VIF) function, in order to select factors suitable for inclusion in the model. It is generally believed that when VIF > 4, variables exhibit multicollinearity. Therefore, explanatory variables that cause multicollinearity are removed before modeling. Based on the VIF test (Table 1), the VIF values for sea surface temperature, sea surface salinity, wind speed at the sea surface wind field, wind direction at the sea surface wind field, flow velocity at the sea surface flow field, flow direction at the sea surface flow field, year, longitude, and latitude were all less than 4. Therefore, these variables were selected to participate in model construction in this study.

In the model fitting process, the stepwise regression method was used to gradually add factors to the model, combined with the Akaike Information Criterion (AIC) [21] and the bias explanation rate to select factors [22]. A lower AIC value and a higher bias explanation rate indicate a better fit of the model [13,23,24]. The iterative process is shown in Table 2. The AIC value for the optimal GAM was 529.056, with a deviation explanation rate of 42.27% and an R^2^ value of 32.47%. The proposed approach incorporates the AD HOC method to address zero values in the data, thus increasing the efficacy of the model when dealing with zero value data. Subsequently, GAM was constructed using the mgcv package in R, and diagnostic factor plots were produced.

To validate whether ln(abundance) follows a normal distribution and whether the standardization using GAM is appropriate, a K-S test was conducted on ln(abundance). The results of the K-S test showed that ln(abundance) tended to follow a normal distribution (μ = 2.12, σ = 1.06) (Figure 2a). Simultaneously, the data points of log(abundance + 1) formed a nearly straight line (Figure 2b), indicating that the assumption of log(abundance + 1) following a normal distribution in this study was reasonable, and using GAM for standardization was appropriate [25].

The significance of the best GAM [26] variable test is shown in Table 3. The F test results indicated that all explanatory variables, including interaction terms, were significant variables and had a significant impact on abundance (*p* < 0.05).

According to AIC, DE, and R^2^, a stepwise regression analysis was conducted to select the variables (YEAR), (LON), (LAT), (SST), and (SSS) as explanatory variables, and (abundance) as the response variable to build the optimal GAM. The expression is as follows:log(abundance + 1) = *s*(YEAR) + *s*(LON) + *s*(LAT) + *s*(SST) + *s*(SSS)(2)

## 3. Results

### 3.1. Spatial and Temporal Changes in Jellyfish Resource Density

#### 3.1.1. Temporal Changes

From 2010 to 2022, the density of jellyfish resources showed periodic fluctuations. There were significant interannual variations in the density of jellyfish resources (*F* = 2.527, *p* = 0.004). Additionally, partial correlation analysis revealed a positive correlation between year and SST (r = 0.358, *p* < 0.001), and a weak correlation with SSS (r = −0.136, *p* < 0.001). Jellyfish resource density was at a relatively high level in 2011 and 2018, while it was at a low level in 2015, 2020, and 2021. The highest density of jellyfish resources occurred at Station 12 in 2011, with a value of 84 ind./h, while there were stations with a density of 0 ind./h in each year. The highest average resource density (10.40 ± 3.20 ind./h) was in 2018, and the lowest in 2015, at 1.60 ± 2.65 ind./h (Figure 3).

#### 3.1.2. Spatial Changes

According to Figure 4, the concentration of high-density jellyfish resources is observed in the nearshore area. During the jellyfish resource density survey conducted in 2013, jellyfish were captured exclusively at coastal sites, while no jellyfish were found at non-coastal sites during that period. Additionally, Station 12 consistently exhibited the highest relative resource density in multiple years, including 2011, 2016, 2017, 2021, and 2022.

When analyzing the average resource density at each site, only Station 12 displayed an average resource density exceeding 10.0 individuals per hour (ind./h). Three sites, namely, Stations 5, 6, and 7, exhibited average resource densities ranging between 5.1 and 10.0 ind./h. Conversely, 11 sites, encompassing Stations 1 to 4, 8 to 11, and 13 to 15, displayed average resource densities between 0.1 and 5.0 ind./h.

### 3.2. GAM Analysis of the Distribution of Jellyfish and Environmental Factors

Figure 5 shows the box plot of explanatory variables and residuals fitted by the best GAM. From this figure, it is evident that the temporal and spatial explanatory variables, namely, year, longitude, and latitude, have residuals that are relatively small and close to zero. Regarding the environmental explanatory variables, the SST has an average residual range of −0.31 to 0.28. Notably, GAM shows the highest positive residual value when the sea surface temperature is 23 °C. Additionally, the residuals of SSS also exhibit fluctuations, with the highest positive residual value occurring at a sea surface salinity of 28‰.

The effect of the year on the density of jellyfish resources shows a decreasing-icreaing-decreasing trend, as depicted in Figure 6. From 2010 to 2015, the density of jellyfish resources gradually declined, reaching its lowest point in 2015. However, between 2015 and 2018, a steady growth trend was observed. However, after 2018, the density decreased again, reaching its lowest point in 2021. Regarding spatial factors, the density of jellyfish resources is relatively evenly distributed in terms of latitude, with minimal variations overall. However, in terms of longitude, a pattern emerges with higher density on the left and lower density on the right. Furthermore, coastal areas exhibit higher densities compared to the open sea.

The impact of sea surface temperature (SST) on the density of jellyfish resources varies across different ranges, making it a significant environmental factor to consider. When the SST ranges between 23 °C and 24 °C, its influence remains relatively stable. However, within the broader range of 20 °C to 26 °C, an overall upward trend is observed. Similarly, the influence of sea surface salinity (SSS) on the density of jellyfish resources shows stability when the salinity ranges between 28.5‰ and 29‰. In contrast, when the salinity falls between 27‰ and 30‰, an initial upward trend is followed by a subsequent downward trend. In conclusion, additional physicochemical and ecological experiments should be conducted to investigate the growth of jellyfish in relation to various environmental factors, such as sea surface temperature (SST) and sea surface salinity (SSS).

### 3.3. The Relationship between Jellyfish Resource Density and Environmental Factors

#### 3.3.1. Ocean Current

The swimming ability of jellyfish is weak, and their distribution is influenced by ocean currents [27]. Pearson correlation analysis was conducted to study the impact of ocean currents on jellyfish. The results showed that there was a strong correlation between the cumulative effect of ocean currents and the site location (r = 0.230, *p* < 0.001), as well as a strong correlation with the longitude within the survey area (r = 0.257, *p* < 0.001). The maximum sea current velocity was 0.515 m/s and the minimum was 0.002 m/s, with an average of 0.110 m/s; the range of sea current directions was broad, mainly between 17.89° and 302.04° (as shown in Figure 7). Based on the spatial distribution of jellyfish resource density in different years, it can be observed that in the years with high jellyfish resource density, the coastal current velocity was higher in 2011 and 2018, while it was lower in 2015.

#### 3.3.2. SST

In this study, the significant impact of sea surface temperature on jellyfish resource density was verified using one-way ANOVA (F = 2.181, *p* < 0.005). A Pearson correlation test revealed a strong negative correlation (r = −0.955, *p* < 0.001) between jellyfish resource density and SST (sea surface temperature). The temperature distribution map depicting the coastal areas surrounding the Shandong Peninsula between 2010 and 2022, as illustrated in Figure 8, reveals substantial variations in yearly temperatures. In 2011 and 2018, the temperature around the monitoring stations was lower, mainly ranging between 20.4 °C and 21.6 °C. In 2012, 2014, 2015, and 2019, the temperature around the monitoring stations was higher, mainly ranging between 21.5 °C and 24.6 °C. In 2011, when the water temperature in the sea around the Haiyang Nuclear Power Plant ranged from 20.4 °C to 21.6 °C, the jellyfish resource density was relatively high; in 2021, when the water temperature in the sea around the Haiyang Nuclear Power Plant ranged from 22.2 °C to 24.6 °C, the jellyfish resource density was relatively low.

#### 3.3.3. SSS

The Pearson correlation test showed that there was a weak correlation (r = −0.058, *p* < 0.005) between jellyfish resource density and sea surface salinity (SSS). The salinity distribution map of the seas surrounding the Shandong Peninsula from 2010 to 2022 (refer to Figure 9) reveals that the salinity levels in the surveyed area typically range between 27.6‰ and 29.4‰. The analysis of the annual variations in jellyfish resource density indicated that the difference in seawater salinity remained insignificant regardless of whether it was a year with high or low jellyfish resource density. Specifically, during the years 2011, 2018, and 2015, the seawater salinity levels in the investigated area predominantly fell within the range of 28.6‰ to 29.2‰. Nonetheless, a comparison of seawater salinity between the years 2018 and 2015 within the region showed some discrepancies. In 2018, the salinity levels at most sampling sites ranged from 28.8‰ to 29‰, and the jellyfish resource density at various sites was at a relatively high level, whereas in 2015, the salinity levels varied between 28.6‰ and 28.8‰ across the majority of sites, and the jellyfish resource density was at a very low level.

## 4. Discussion

### 4.1. The Spatiotemporal Distribution Characteristics of Jellyfish Resources

The resource density of jellyfish showed significant differences between 2010 and 2022. The jellyfish resource density was at a relatively high level in 2011 and 2018, while it was at a low level in 2015, 2020, and 2021. The highest resource density was observed in 2011, with an average of 84 ind./h at site 12. And the highest average resource density was recorded in 2018. This indicates that the resource density of jellyfish fluctuates across different years. The interannual differences are significant, and the main influencing factors include release time, sea temperature, food availability, ocean currents, and natural disasters, among others [12]. Climate change can profoundly affect marine ecosystems through alterations in seawater temperature and modifications in ocean circulation. For example, the periodic variations in jellyfish resource density may be influenced by El Niño. From 2010 to 2022, the occurrence of El Niño could lead to the migration or weakening of nutrient-rich cold upwelling currents, causing changes in sea salinity, temperature, and oxygen levels [28,29]. These changes can result in a decline in the quantity of algae and zooplankton, the primary food source for jellyfish, ultimately affecting the fluctuations in jellyfish resource density.

This study revealed significant variations in jellyfish resource density among different locations, with a higher concentration found in coastal waters. Several factors can contribute to this phenomenon. During their outbreak period, jellyfish have a high demand for food, preying on zooplankton, fish eggs, and larvae, thereby competing for food with fish. Large outbreaks of jellyfish can regulate the numbers of other zooplankton and fish larvae through top-down control mechanisms [30]. Additionally, through the nutrients produced by their own metabolism, jellyfish can influence the population size of phytoplankton through bottom-up control mechanisms or trophic cascades [30]. Firstly, coastal areas receive greater nutrient inputs, including river inflows [31], and water pollution. This increases the abundance of plankton and other benthic organisms, providing a richer food source for jellyfish. Additionally, coastal areas provide more favorable habitats for jellyfish growth and reproduction, such as shallow waters, reefs, and shallows, which create ideal environments for both benthic and planktonic organisms, leading to higher concentrations of jellyfish reproduction. The interaction between land and sea in coastal areas results in more stable water temperature and salinity. Moreover, these areas exhibit complex and diverse aquatic ecosystems with greater biodiversity and interactions, making it more likely for jellyfish to find suitable ecological niches for survival. Station 12 had the highest observed resource density, with an average of 84 individuals per hour. This can be attributed to its proximity to the river mouth, which introduces abundant nutrients, enhances primary productivity, and enriches plankton and other benthic organisms, providing a richer food source for jellyfish. This corresponds with the findings of WANG Bin: the relatively high temperatures and low salinity areas of coastal estuaries provide an optimum environment for the growth of juvenile medusae of *Nemopilema nomuraiin* [32]. Previous studies have shown that scyphozoan jellyfish appear in offshore areas during spring and migrate to estuarine habitats for sexual reproduction in summer [33].

### 4.2. Analysis of GAM

The Generalized Additive Model (GAM), as an extension of the Generalized Linear Model (GLM), can use spline smoothing functions to add more Taylor series terms (i.e., polynomial terms), making the model a nonlinear model. This allows handling the nonlinear relationship between the response variable and multiple explanatory variables. GAM has been widely used in research on fishery resources [33,34,35]. In this study, based on the survey data of resources around the Haiyang Nuclear Power Plant in the years 2010–2022 from Chinese trawl fishing vessels and marine environmental data provided by Copernicus Marine Data, the impact of environmental factors on the density of jellyfish resources is studied using GAM.

GAM was tested for statistical distribution, and the log(abundance + 1) tended to follow a normal distribution, validating the rationality of using GAM for standardization. The selected optimal GAM accounted for 42.2% of the variation in abundance, indicating a certain degree of predictive capability to elucidate changes in abundance data. The remaining 57.8% of the variance could be attributed to other factors or errors, highlighting the potential need to incorporate additional explanatory variables such as the sea surface wind field and ocean current. All explanatory variables, including interaction terms, exhibited significant effects on abundance. Notably, the statistical significance of sea surface temperature (SST) and sea surface salinity (SSS) exceeded that of the other explanatory variables, thereby implying that SST and SSS contribute more significantly to the density of jellyfish resources. Although these explanatory variables exert diverse effects on the density of jellyfish resources, interactions among them exist. For instance, variations in sea surface temperature across different years can influence jellyfish growth, reproduction, and migration. Elevated temperatures may foster jellyfish growth and reproduction, leading to increased resource density, while excessively high or low temperatures may have detrimental effects on jellyfish survival, resulting in a decrease in resource density. Alterations in SSS can impact the survival and reproductive conditions of jellyfish, consequently generating distinct spatial distributions. These interactions between factors such as time, temperature, and space enable a comprehensive analysis of the seasonal and geographical distribution pattern.

In the optimal GAM, temporal and spatial explanatory variables exhibit smaller residuals in the box plot, while environmental explanatory variables such as sea surface temperature and salinity show fluctuations in residuals. The average residual range for SST is −0.31 to 0.28 °C, with the highest positive residual value occurring at a sea surface temperature of 23 °C. The residuals for SSS also exhibit some fluctuations, with the maximum positive residual value observed at a sea surface salinity of 28‰. The impact of SST on the density of jellyfish resources remains relatively stable when the sea surface temperature ranges from 23 °C to 24 °C, generally increasing within the range of 20 to 26 °C. Similarly, the impact of SSS on the density of jellyfish resources remains relatively stable between a salinity of 28.5 and 29‰. However, within the salinity range of 27 to 30‰, SSS demonstrates an increasing trend followed by a decreasing trend.

Various natural factors (sea surface temperature, sea surface salinity, tidal front, surface water flow, turbidity, and dissolved oxygen) and anthropogenic factors (water quality degradation, overfishing, translocation, and habitat changes) play a key role in triggering jellyfish outbreaks and aggregations [23,36,37,38,39]. Ocean warming generally leads to an increase in jellyfish density, and the global ocean temperature rise may alter the distribution of certain jellyfish species, with high sea temperatures favoring jellyfish outbreaks [40], while low sea temperatures may inhibit jellyfish proliferation. The growth of polyps and ephyrae is mainly influenced by temperature [41], and increasing temperatures may enhance polyp differentiation, while the presence of low temperatures likely creates unfavorable conditions for the survival of *Nemopilema nomurai*, even within a eutrophic environment [42]. Additionally, the degradation of water quality, increased turbidity, and diminished dissolved oxygen levels can result in scattered aggregation and swarms [43].

### 4.3. The Impact of Environmental Factors on Jellyfish Aggregation

#### 4.3.1. Ocean Current and Wind Field

Jellyfish resource density is related to the direction of ocean currents and sea winds. Jellyfish have weak swimming capabilities, and their distribution is influenced by the transportation of ocean currents. Many scholars have confirmed through research the impact of ocean currents and sea winds on the density of jellyfish resources [27,43,44,45]. The direction of ocean currents and sea winds can affect the distribution and aggregation of jellyfish, influencing their drifting and migration. Sea winds can cause jellyfish to aggregate near coastlines or specific areas. If the direction of ocean currents aligns with the environmental requirements of the jellyfish life stage, the jellyfish population will increase. On the other hand, if the ocean currents help carry jellyfish away from their breeding or feeding areas, the jellyfish population may decrease. Additionally, sea winds can generate waves and water currents, which may impact the movement and aggregation of jellyfish.

Based on data analysis, it is evident that there is a significant correlation between the cumulative impact of ocean currents and the spatial distribution of monitoring stations (r = 0.230, *p* < 0.001), as well as a strong association with the longitudinal coordinates within the surveyed region (r = 0.257, *p* < 0.001). The pronounced influence of nearshore ocean currents plays a pivotal role in the clustering of high-density jellyfish populations along the coastline. As such, it is hypothesized that the interplay of wind dynamics, shelf currents, and topographic characteristics serves as a key determinant in shaping the jellyfish distribution patterns. The atmospheric and oceanic conditions in July create a conducive environment for jellyfish aggregation. The dominant onshore winds and the cumulative impact of currents actively contribute to the congregation of jellyfish along the shoreline. However, according to the correlation results, the effect of sea field on the density of jellyfish resources was small, thus proving that sea field was not an important factor affecting the variation of jellyfish resource density.

#### 4.3.2. SST and SSS

Xupeng Chi et al.’s study indicates that polyps have an optimal temperature tolerance of 20 °C. Under natural conditions [46,47], jellyfish growth thrives at 18.78 °C [48,49], and once the sea surface temperature surpasses 19.5 °C, the jellyfish transition from the polyp to the medusa form [50]. Jellyfish cease movement when the temperature drops below 3 °C or exceeds 35 °C [51]. From Figure 9, it can be seen that in years with high jellyfish resource density such as 2011 and 2018, the sea surface temperature is approximately between 20.4 °C and 21.6 °C; in years with low jellyfish resource density such as 2021, the sea surface temperature is approximately between 22.2 °C and 24.6 °C. Preliminary analysis indicates that the optimal temperature tolerance of jellyfish is around 20 °C. This is consistent with previous research results. Studies have shown that the nearshore Kuroshio path can affect the distribution of jellyfish [52]. Based on this, we conducted a literature search and found that in the summer of 2011, under the influence of the Kuroshio, there was a strong thermocline, which maintained a stable cold water mass with low temperature, high salinity, and rich nutrients in the central Yellow Sea [53]. This finding aligns with our study’s conclusion that the surveyed area in July 2011 was in a state of low temperature and high salinity compared to other years. Therefore, it is reasonable to infer that the anomalous conditions in the surveyed area in July 2011 were caused by the influence of the Kuroshio, which in turn led to an increase in jellyfish resource density.

The osmoregulation ability of jellyfish allows them to tolerate variations in salinity by maintaining internal saltwater balance through controlling salt absorption and excretion [54]. Jellyfish adjust their ionic concentration to that of the surrounding seawater and tend to remain in areas with consistent salinity levels, resulting in localized accumulation [55]. Thus, jellyfish can survive in areas with low salinity in river estuaries, high salinity in salt fields, and the open sea. However, the survival rate of jellyfish significantly decreases at a salinity of 40‰. While the polyp stage demonstrates a strong adaptive capacity to changes in external salinity due to their cells possessing robust osmoregulatory abilities, the medusa and adult stages are comparatively less adaptive [56]. Therefore, further freshwater training can be employed in jellyfish cultivation processes. From 2010 to 2022, the salinity in the waters surrounding the Haiyang Nuclear Power Plant has remained stable between 27.4 and 30.2‰, within the salinity tolerance range of jellyfish. Therefore, it provides suitable salinity conditions for the survival of jellyfish.

## 5. Conclusions and Prospects

This study utilized the Generalized Additive Model (GAM) to investigate the distribution patterns of jellyfish in the southern sea area of the Shandong Peninsula and their correlation with environmental factors. The findings indicated that the key variables influencing jellyfish resource density included year, longitude, latitude, sea surface temperature (SST), and sea surface salinity (SSS). Building upon these outcomes, a detailed analysis was conducted to assess how these factors impact the abundance of jellyfish resources. Moreover, the study highlighted the significant role of ocean currents in shaping the density of jellyfish resources. A comprehensive examination of the interplay between ocean currents and jellyfish resource density revealed that the environmental conditions surrounding the Haiyang Nuclear Power Plant provided a conducive environment for jellyfish survival. Then, we explored how El Niño and the Kuroshio influence jellyfish resource density by affecting environmental factors such as SSS (sea surface salinity) and SST (sea surface temperature). By examining the interplay between jellyfish outbreaks and environmental factors, we gained insights into the intrinsic and pivotal external triggers of these outbreaks. Drawing from our research findings, it is advisable for nuclear power plants to strategically deploy pertinent environmental monitoring sensors and equipment, and to develop a tailored environmental impact model focusing on the distribution patterns of obstructing species [56]. Furthermore, collaboration between nuclear power plants, maritime authorities, local governmental bodies (comprising meteorology, hydrology, maritime affairs, fisheries, etc.), research institutions, professional entities, and local fishermen is essential to establish a cohesive defense mechanism [57]. Should the need arise, efforts with nearby fishermen should be coordinated to promptly address outbreak occurrences at sea.

The research findings suggest that utilizing GAM to assess the influence of environmental factors on jellyfish is viable, elucidating the impact of variables such as temperature and salinity on jellyfish outbreaks. However, based on the outcomes of model testing, there remains scope for enhancing the model’s explanatory capability. Hence, future investigations will explore the integration of additional external factors affecting jellyfish outbreaks, such as eutrophication and the influence of El Niño on jellyfish resource density. Furthermore, in the vicinity of the Haiyang Nuclear Power Plant, there has been a notable increase in the occurrence of *Sargassum horneri* outbreaks in recent years, posing a potential threat to the safety of the plant’s cooling system. Consequently, there will be a concerted effort to bolster the assessment of *Sargassum horneri* resources to fortify the nuclear power plant against marine biological hazards.

## Figures and Tables

**Figure 1 biology-13-00433-f001:**
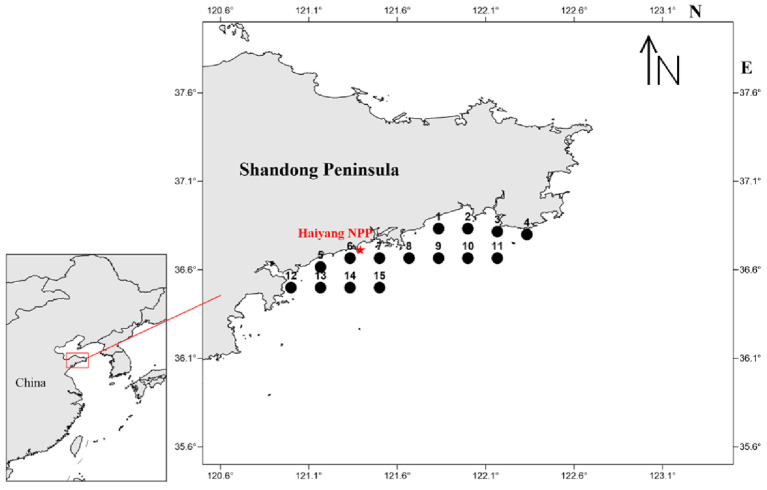
Distribution of jellyfish resource investigation stations in the seawater intake area of the Haiyang Nuclear Power Plant.

**Figure 2 biology-13-00433-f002:**
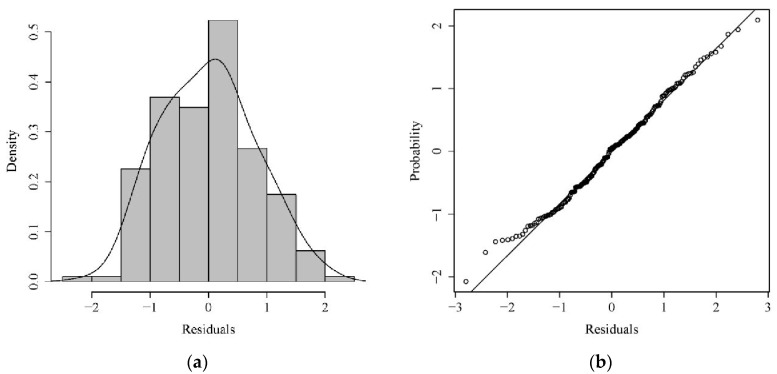
The frequency distribution (**a**) and its test (**b**) of ln(abundance) of jellyfish in the southern sea area of Yantai and Weihai, Shandong, from 2010 to 2022.

**Figure 3 biology-13-00433-f003:**
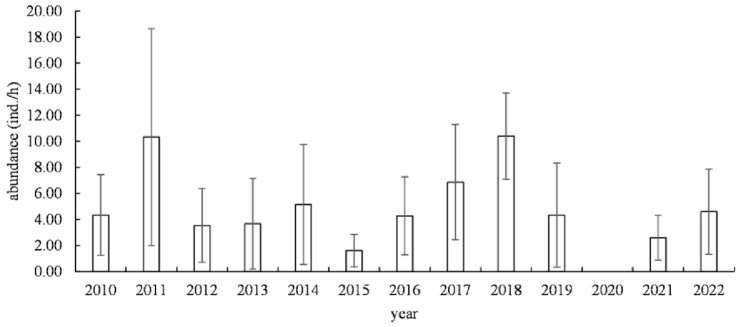
Interannual variation in jellyfish resource density in the southern sea areas of Yantai and Weihai, Shandong Province, from 2010 to 2022.

**Figure 4 biology-13-00433-f004:**
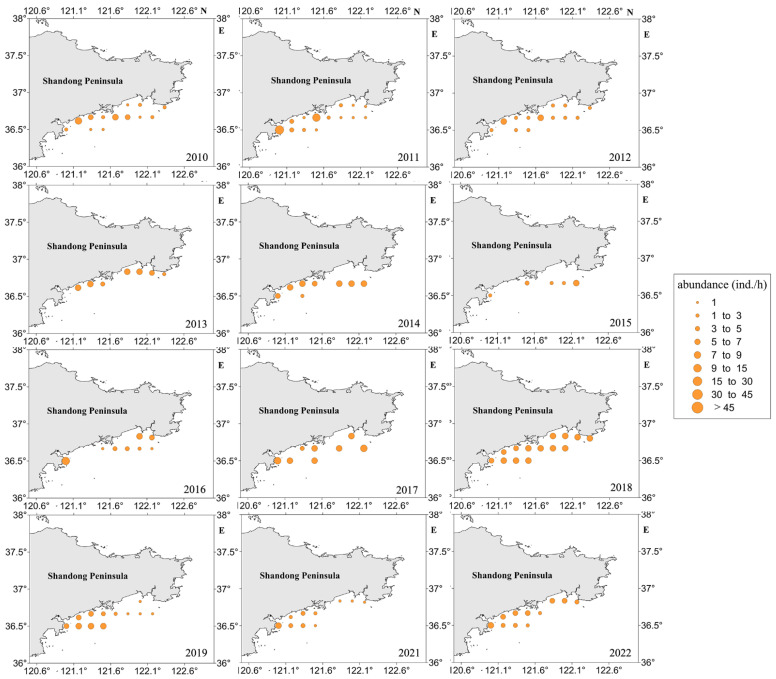
Density distribution map of jellyfish resources in the sea area around Haiyang Nuclear Power Plant in Shandong Province, China, from 2010 to 2022.

**Figure 5 biology-13-00433-f005:**
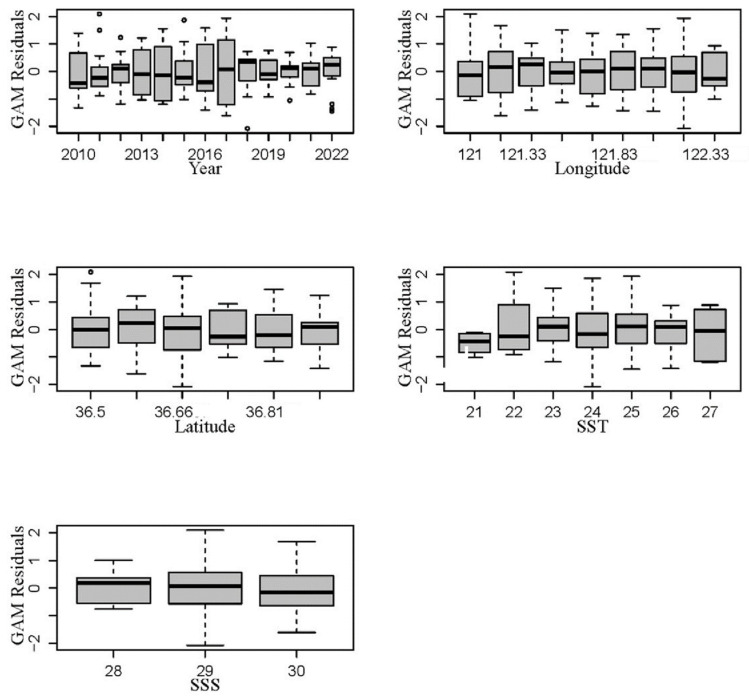
Box plot of explanatory variables and residuals fitted by the best GAM. (The black dots in the diagram represent outliers in the respective factors).

**Figure 6 biology-13-00433-f006:**
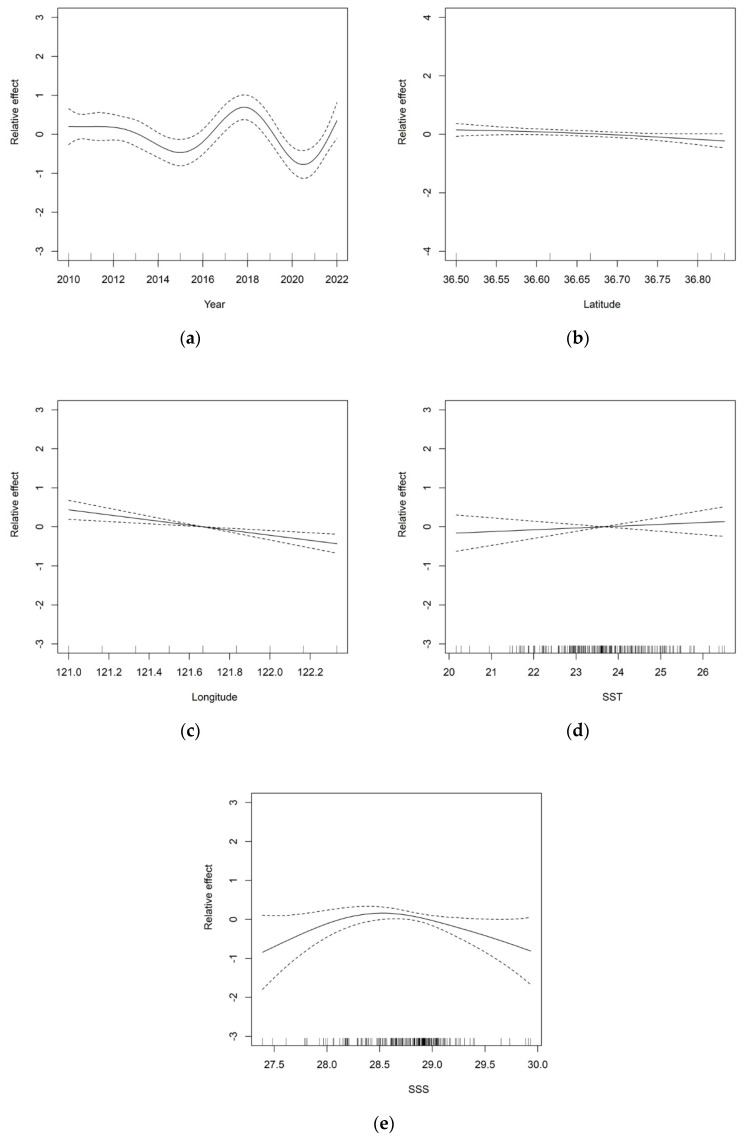
The impact of each explanatory variable on the density of jellyfish resources in the southern waters of Yantai and Weihai, Shandong Province, from 2010 to 2022: (**a**) year; (**b**) latitude; (**c**) longitude; (**d**) SST; and (**e**) SSS. (The dashed lines in the diagram represent the 95% confidence interval, and the solid lines represent the actual values).

**Figure 7 biology-13-00433-f007:**
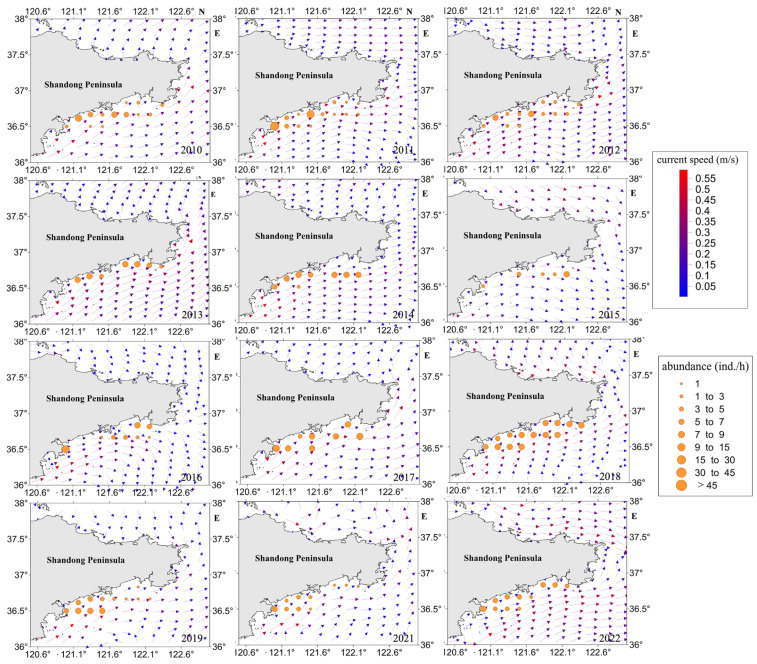
Overlay map of sea currents and jellyfish resource density in the Shandong Peninsula from 2010 to 2022.

**Figure 8 biology-13-00433-f008:**
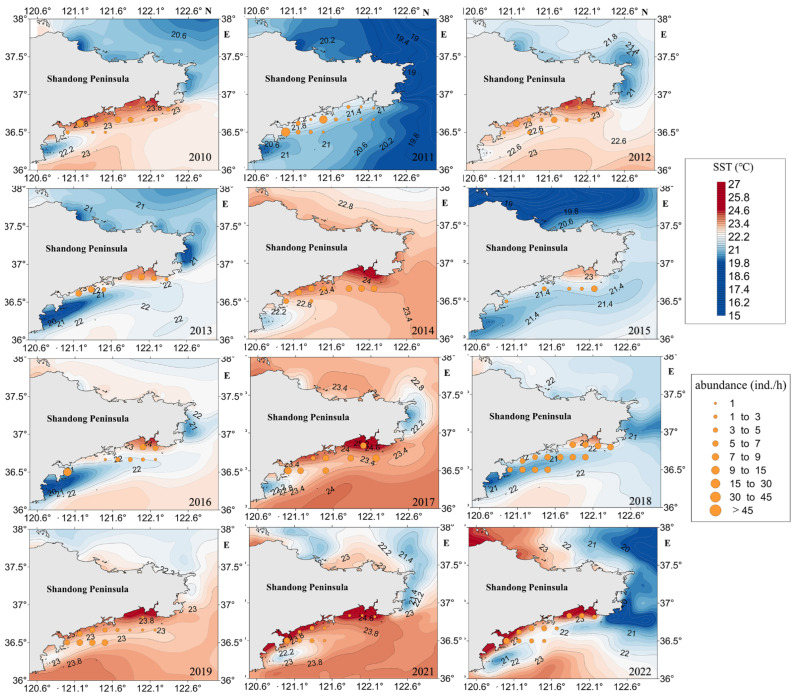
Overlay map of sea surface temperature (SST) and jellyfish resource density in the Shandong Peninsula from 2010 to 2022.

**Figure 9 biology-13-00433-f009:**
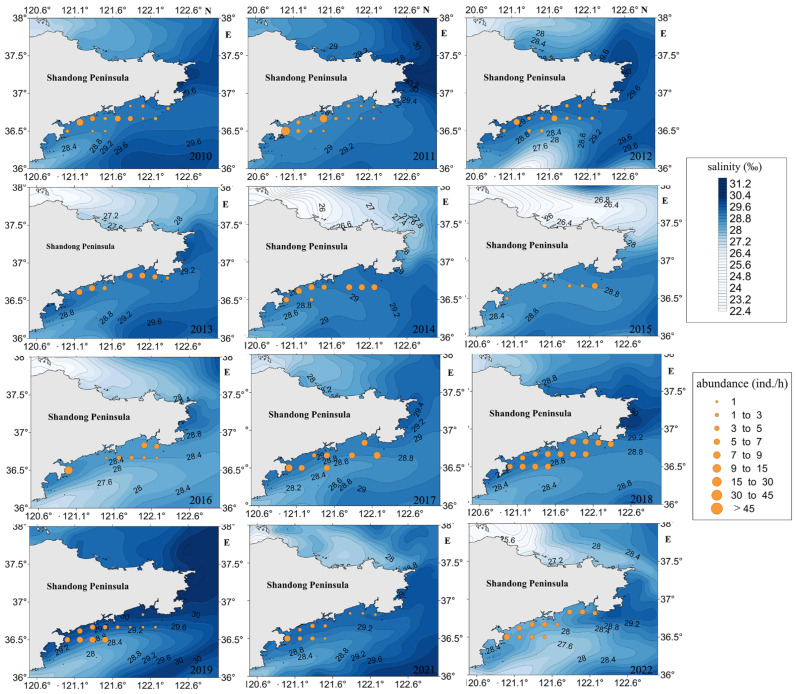
Overlay map of SSS and jellyfish resource density in the Shandong Peninsula from 2010 to 2022.

**Table 1 biology-13-00433-t001:** Variable VIF testing results.

Variable	YEAR	LON	LAT	SST	SSS	SSC__speed_	SSC__dir_	Wind__speed_	Wind__dir_
VIF value	1.52	2.85	3.4	1.94	1.22	1.71	1.64	1.19	1.21

**Table 2 biology-13-00433-t002:** GAM selection based on AIC.

GAM	R^2^ (%)	AIC	Explanation Rate (%)
Log(abundance + 1) = *s*(YEAR)	14.39	550.6193	26.85
log(abundance + 1) = *s*(YEAR) + *s*(LON)	20.81	542.8741	31.77
log(abundance + 1) = *s*(YEAR) + *s*(LON) + *s*(LAT)	21.59	539.7905	32.67
log(abundance + 1) = *s*(YEAR) + *s*(LON) + *s*(LAT) + *s*(SST)	27.33	534.9276	37.39
log(abundance + 1) = *s*(YEAR) + *s*(LON) + *s*(LAT) + *s*(SST) + *s*(SSS)	32.47	529.0556	42.27
log(abundance + 1) = *s*(YEAR) + *s*(LON) + *s*(LAT) + *s*(SST) + *s*(SSS) + *s*(Wind__speed_)	32.11	530.8148	41.96
log(abundance + 1) = *s*(YEAR) + *s*(LON) + *s*(LAT) + *s*(SST) + *s*(SSS) + *s*(Wind_speed) + *s*(Wind__dir_)	29.77	532.7855	40.33
log(abundance + 1) = *s*(YEAR) + *s*(LON) + *s*(LAT) + *s*(SST) + *s*(SSS) + *s*(Wind_speed) + *s*(Wind__dir_) + *s*(SSC__speed_)	30.69	531.2447	41.23
log(abundance + 1) = *s*(YEAR) + *s*(LON) + *s*(LAT) + *s*(SST) + *s*(SSS) + *s*(Wind__speed_) + *s*(Wind__dir_) + *s*(SSC__speed_) + *s*(SSC__dir_)	27.51	533.1651	39.84

**Table 3 biology-13-00433-t003:** ANOVA of the optimal GAM.

Parameter	df	F	*p* Value
factor(YEAR)	12	5.819337	2.28 × 10^−8^
factor(LON)	8	3.512305	8.91 × 10^−3^
factor(LAT)	5	4.668616	5.16 × 10^−3^
*s*(SST)	3	0.787265	0.416
*s*(SSS)	3	2.754958	0.049

## Data Availability

The original contributions presented in the study are included in the article, and further inquiries can be directed to the corresponding authors.

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
