# Peer review of "Analysis of the Distribution Characteristics of Jellyfish and Environmental Factors in the Seawater Intake Area of the Haiyang Nuclear Power Plant in China"

_biology, 2024, doi:10.3390/biology13060433_

Round 1
Reviewer 1 Report
Comments and Suggestions for Authors
The manuscript (MS) was designed to show long term data report of jellyfish density in a large area of a power plant including intake water system. According to the title of MS, the density would be expected to have always high density of the jellyfish regardless of years. Indeed the there would have been no annual differences if the power plant affected the jellies distribution. Therefore, the distribution could be related to the atmospheric index interaction and availability of food (zooplankton) of the jellies besides the limited minor physical effect on it.
The Introduction could be written more compressively including zooplankton distribution as food of jellies and the area could be affected by El Nıno or/and pacific atmospheric index where there are some atmospheric indices rather than power plant effect if such parameters were not measured during the present study.
Material and Methods In Matlab what is the command to solve GAM?
Hauling or towing were made for jelly collection. If vertical haul, give the depth, if the horizontal towing give the layered depth and diel vertical distribution of jellies. Data collection was monthly or once a time in a year during the long term data. Data were collected at same months of each year and so on many questions could be generated.
If there is normality test for data, give results in Mat and Met instead of Results. The formula of GAM seemed to be GLM's by deriving colinearity.
Statistical analyses could be detailed priory for better understanding until the readers reach the results
Results and Analysis is not convenient for the research article.
Only results is enough
Fig. 2 showed a periodical increase and decrease of jellies density. For effect of power plant this interannual difference was inherently not expected if the authors specified the title on the power plant. This could be due to pacific indices 7-8 yeas fitted with the Pacific indices and El Nino.
Station 12 not always to have high density in Fig. 3 why stat 12 is considered priory.
3.2.1 and some parts of 3.2.2 could move to Mat and met.
GAM or GLM could seem to be applied partially (Table 2) instead of non-partial GAM.
Plot also density across the environmental data besides the residuals and give spearman correlation.
Give statistical results instead of surfer plot of density with each physical parameters. Do not need to plot each variable of physics. Pls elucidate how to match inter-annual physical parameters measurements with the density of jellies if there are no seasonal measurements in each year.
Jellies like eutrified region where the food (zooplankton) is variable. Zooplankton is missing in this study.
From the distribution map, the wind stress seemed to affect the jellies distribution rather than other physics. That's why the map alone makes non-sense to understand.
Pearson or spearman correlation could be applied to the data, but partial correlation including all set of physical data versus the density data to figure out hidden variables of the environmental parameters.
Discussion could be rewritten after revision was made according to comments above.
Comments on the Quality of English LanguageSome syntax errors are available in the words discretely
Reviewer 2 Report
Comments and Suggestions for Authors
Very interesting study. Some of the suggestions are below,
It would be nice to include ‘in China’ in the title to make the location clearer.
Abstract: Mention about the future potential very briefly in the end (How it will be done and how it will be advantageous in a bigger picture)
Introduction:
1. How this Shandong Haiyan Nuclear Plant is important and why its important to protect it? I understand the manuscript is more about general nuclear plants, but it would be nice to give some information about this power plant.
2. More information but less references included in this section
3. Why was Shandong Peninsula chosen?
4. How this analysis will be done (methodology) should be included here briefly
Results and Analysis:
1. Figure 3, 7, 8, 9 texts are difficult to read especially the legend description
Conclusion:
1. Any measures/actions that need to be taken by the government or anybody else? What next step needs to be taken without disturbing the natural ecosystem much?
2. I feel there is no good end to this conclusion
3. If possible, write a separate future prospective section
4. What about other marine organism apart from jellyfish?
5. Could there be any other significant environmental factors involved?
Figure formatting and some minor edits need to be done throughout the manuscript
(eg. or-ganism?)
Reviewer 3 Report
Comments and Suggestions for Authors
Review for the paper "Analysis of the distribution characteristics of jellyfish and environmental factors in the seawater intake area of Haiyang Nuclear Power Plant" by Yunpeng Song, Tiantian Wang, Minsi Xiong, Jie Ying, Yongchuang Shi, Guoqing Zhao, Xiumei Zhang, Xiaodan Liu, Cankun Lin, Zuli Wu, Yumei Wu submitted to "Biology".
General comment.
Jellyfish have key characteristics that place them in an influential position to control energy flows through pelagic food webs, such as high reproduction and growth rates, broad planktivorous diets, and relatively few apparent predators. Jellyfish are important consumers of zooplankton and can strongly influence energy and carbon pathways through food webs when their abundance is high. Jellyfish can negatively impact fisheries by competing with zooplanktivorous fish, preying on fish eggs and larvae, and indirectly competing with higher trophic levels by reducing plankton available to planktivores. In addition, medusae outbreaks can have many negative consequences for coastal industries. In particular, the intake systems of nuclear power plants can be clogged with jellyfish, which can be a potential threat to their stable exploitation. The authors conducted a survey in an area where there is a water intake for the cooling water system of the Haiyang Nuclear Power Plant. They also correlated the density of medusae with a number of environmental variables and found that temperature and salinity were the main factors. Although the topic is interesting, there are major flaws that need to be addressed in the revised version.
Major points.
1. Introduction. The authors must provide a broader overview of the problem. The most cited papers in the introduction are Chinese papers. Therefore, other international research must be included.
2. Introduction. The composition and occurrence of the main jellyfish taxa inhabiting the coastal waters must be included in the paper.
3. Introduction. The novelty of the study is unclear. There are a number of studies dealing with various aspects of jellyfish biology and ecology in the coastal waters of China. What is the research gap that the present study fills?
4. Materials and Methods. I suggest including a brief section describing the general environmental conditions in the region, such as climate, circulation, bathymetry, and currents. This information would enhance the reader's understanding of the context in which the research was conducted.
5. Materials and Methods. The choice of summer season must be clearly explained in the MS. The authors should present literature data for other seasons to indicate the seasons of maximum medusa abundance in the region.
6. Materials and Methods. The procedure for collecting and calculating jellyfish CPUE must be described in more detail (depth of trawling, time of trawling, etc.).
7. Results. When describing the environmental conditions (sections 3.3.1-3.3.4), it is important to compare the results using relevant statistical criteria such as ANOVA, Kruskal-Wallis test or other appropriate methods. This will help to assess the significance of the observed differences.
8. Discussion. The role of other factors (e.g. interspecific competition for food with macrozooplankton and food, top-down control by higher consumers, etc.) must be thoroughly discussed.
9. Discussion. The authors pointed out that the richest jellyfish populations occurred in the coastal waters affected by the river discharge. What about water pollution? Pollutants are considered to be one of the most important factors controlling plankton abundance in nearshore environments.
10. To make the study more attractive to an international audience, it would be informative to compare the results with similar ecosystems. This comparative analysis will deepen the study and emphasize its relevance beyond the immediate research area.
11. Discussion. It is proposed to provide key recommendations to coastal managers and officials dealing with the nuclear power industry for rational operations in the coastal zone during jellyfish outbreaks.
Specific remarks.
L59 onwards. Latin names of species and genera must be in italics. L395.
Figure 2 caption. What do the vertical bars mean?
Figures 3, 7-10. Please increase the font size in all figures.
Section 3.2.1. I think this needs to be removed from the main text and moved to the Supplementary Data.
Section 3.2.2 (L200-210 with Table 1) contains redundant information (so does the Material and Methods) and can be moved to the Supplementary Data.
L206. R2. "2" must be in the upper index.
L319. Consider replacing "bait" with "food availability".
Reference List. Please carefully check and correct the Latin names of medusae and other animals. These must be italicized throughout the reference list.
Comments on the Quality of English LanguageThe English is good.
Round 2
Reviewer 1 Report
Comments and Suggestions for Authors
The manuscript was improved to be a publication in the journal.
The MS could be accepted after editorial check.
Reviewer 3 Report
Comments and Suggestions for Authors
No further comments.